# Sub-Milliwatt Transceiver IC for Transcutaneous Communication of an Intracortical Visual Prosthesis

**Adedayo Omisakin** *,† , **Rob Mestrom** †, **Georgi Radulov** † and **Mark Bentum** †

Department of Electrical Engineering, Eindhoven University of Technology,
5600 MB Eindhoven, The Netherlands; r.m.c.mestrom@tue.nl (R.M.); g.radulov@tue.nl (G.R.);
m.j.bentum@tue.nl (M.B.)

**\*** Correspondence: a.e.omisakin@tue.nl; Tel.: +31-6515-49995

**†** These authors contributed equally to this work.

**Abstract:** An intracortical visual prosthesis plays a vital role in partially restoring the faculty of sight in visually impaired people. Reliable high date rate wireless links are needed for transcutaneous communication. Such wireless communication should receive stimulation data (downlink) and send out neural recorded data (uplink). Hence, there is a need for an implanted transceiver that is low-power and delivers sufficient data rate for both uplink and downlink. In this paper, we propose an integrated circuit (IC) solution based on impulse radio ultrawideband using on-off keying modulation (OOK IR-UWB) for the uplink transmitter, and binary phase-shift keying (BPSK) with sampling and digital detection for the downlink receiver. To make the solution low-power, predominantly digital components are used in the presented transceiver test-chip. Current-controlled oscillators and an impulse generator provide tunability and complete the on-chip integration. The transceiver test-IC is fabricated in 180 nm CMOS technology and occupies only 0.0272 mm$^2$. At 1.3 V power supply, only 0.2 mW is consumed for the BPSK receiver and 0.3 mW for the IR-UWB transmitter in the transceiver IC, while delivering 1 Mbps and 50 Mbps, respectively. Our link budget analysis shows that this test chip is suitable for intracortical integration considering the future off-chip antennas/coils transcutaneous 3–7 mm communication with the outer side. Hence, our work will enable realistic wireless links for the intracortical visual prosthesis.

**Keywords:** implanted transceiver; impulse radio ultrawideband (IR-UWB); link budget; low-power; intracortical visual prosthesis; neural recording; non-coherent digital demodulator

## 1. Introduction

Visual impairment is a sensory challenge with a significant impact on the daily life of patients. Nearly 216 million people worldwide are visually impaired, of which 36 million are currently blind [1,2]. For a lot of these blind people, stimulating the visual cortex is the treatment of last resort due to damage to the visual pathway. Such a system, an intracortical visual prosthesis, consists of implanted electrodes, a signal processor on the outside as well as a camera and a feedback loop. [3]. In the context of this work, the Dutch NESTOR project (NEuronal STimulation fOr Recovery of function) aims to implant 1024 electrodes for stimulation and recording in the brain [4,5] (funded by the Netherlands Organization for fundamental research—NWO). Wireless connection between the implanted electrode and the external processor is desired to facilitate mobility and to avoid infections during long-term use [6]. On the implant side, this wireless connection involves uplink (for recording), downlink (for stimulation) and wireless power transfer.

This work focuses on the implanted transceiver IC, which is the transmitter for sending out of the head (uplink) and the receiver for receiving stimulation data (downlink).

There are a few possible configurations for the wireless system. The implanted electrodes are grouped into arrays of 64 electrodes. Our approach used in this work is to place the central transceiver beneath the skin. The electrode arrays are tethered to the central transceiver [7]. Figure 1 illustrates our solution. Our approach is more easily scalable

than placing an implanted transceiver on each electrode array beneath the skull, making them independent [8], but requiring multiple implanted transceivers. For instance, our 1024 electrodes are implemented by 16 arrays of 64 electrodes (typical of Utah arrays). Taking an approach with an independent transceiver per electrode array would require 16 transceivers in total. Our proposed tethered approach, which requires only one central transceiver as in Figure 1, is more scalable. Furthermore, in our approach, a single transceiver unit could be used with a smaller or larger amount of tethered electrode arrays. For the concept we propose, this does not matter. The electrode arrays need to be placed at the desired location, while communication with the outside world will not change, due to the central unit. Our approach is preferred over using a central transceiver placed beneath the skull, which will face more attenuation and path-loss by the bone tissue (up to 10 dB more) [9]. The main challenge in our tethered solution is the possible micro-motion of the implanted connecting wires; this can be partly alleviated with better packaging and implantation. In this paper, we focus on the implanted transceiver IC design and analyze its link budget in the context of a complete system.

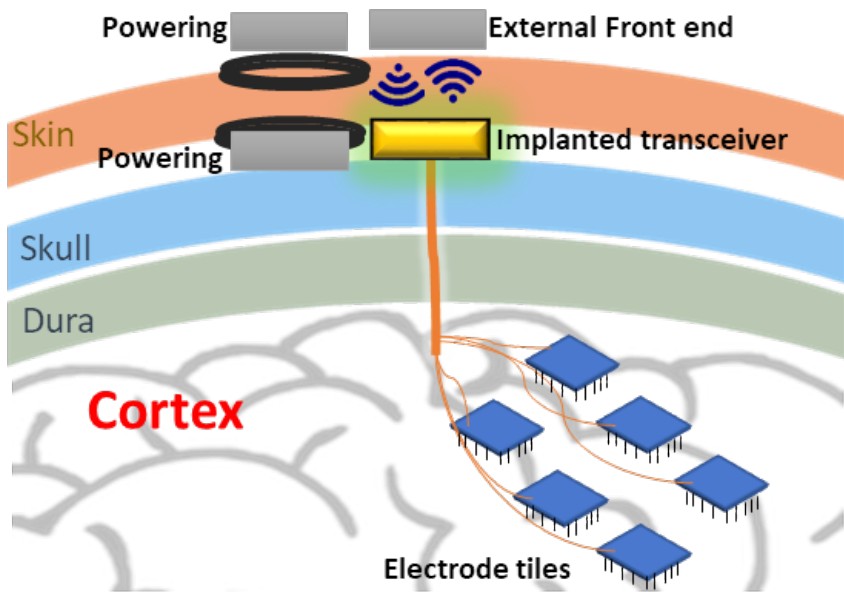

**Figure 1.** System layout of the wireless enabled visual prosthesis. Note: The transmitter and receiver, without the coil/antenna, is covered in this paper.

Generic biomedical telemetries have been proposed for the implanted transmitter for recording and also for an implanted receiver for stimulation. However, in this work, we combine them into a single implanted transceiver for bi-directional communication in an intracortical visual prosthesis. In [10], optical communication was attempted for the uplink. However, this faces a challenge of alignment sensitivity. Impulse radio ultrawideband (IR-UWB) was used in [11], showing a promising result. A high data rate was achieved in [12] using an IR-UWB transmitter. In our work, for the recording, we use IR-UWB for the uplink because of its low-power and high data rate potential at its transmitter, which will be at the implant side. For our implanted transmitter, we applied current control to make the impulse generator tunable for pulse width. A straightforward on-off keying modulation (OOK) scheme is implemented, based on a simple D-latch to achieve low-power consumption. Furthermore, a current-controlled variable oscillator is proposed to tune the number of pulses per bit. The transmitter is built using predominantly digital components to strive towards low power consumption while delivering a high data rate (a minimum of 23 Mbps is required [13]). The potential of digital-based designs is discussed in [14,15], which is beneficial to low-power transceivers.

For the downlink receiver, an analog differential phase-shift keying (DPSK) receiver was used in [16]. Pulse delay modulation (PDM) was used in [17] and an Amplitude Shift Keying (ASK) to Frequency Shift Keying (FSK) conversion receiver in [18], both for low power consumption. However, for our implanted receiver we took an approach of a non-coherent digital receiver, with low-carrier frequency (4 MHz) and using an inductive link. The implanted receiver's low-power and sufficient data rate due to its predominantly digital components and low frequency are the reasons for its selection. The receiver is above the minimum data rate requirement of 200 kbps for the downlink [19].

In this work we describe the design of our fabricated implanted transceiver (excluding coils and antenna) on IC, based on CMOS 180 nm technology. Our idea is to: (1) use predominantly digital components to achieve lower power consumption; (2) provide tunability for the implanted IR-UWB transmitter with current control for its oscillator and impulse generator; and (3) make the implanted receiver adjustable with current controls to optimize for an inductive link.

## 2. System Requirements and Considerations

For the implanted transceiver, which comprises of the transmitter for uplink (sending out data from the implanted side) and its receiver for downlink (receiving stimulation data), the following essential system requirements are considered:

1.  Power consumption: Based on power consumption reported in [11,20], the projected power consumption of the implant side of the 1024 electrode visual prosthesis without a wireless interface is in the order of 100 mW. Considering the wireless power transfer, and possible battery constraints at the implant side, it is desired that the wireless system adds no more than 10–30% extra power to the power budget at the implant side. This implies that only a low-power solution for the implanted transceiver will create a viable system. The external transceiver above the skin will be allowed to consume more power. Our proposed non-coherent BPSK receiver will fulfill this due to its simple architecture and predominantly digital components. For the uplink, on-off keying (OOK) is used for the IR-UWB transmitter because of its simpler architecture making for a low-power transmitter.

2.  Transmission data rate: It is required to transmit a minimum of 23 Mbps for compressed data with electrode recordings [13]. The receiver is required to handle a minimum of 200 kbps of stimulation data [19]. IR-UWB for the uplink has the potential to deliver high data rate for short distances. Using an inductively-coupled link with BPSK at low frequency will provide a sufficient data rate, while yielding a low-power solution [21].

3.  Bit-error-rate (BER): the uplink and downlink system should give a BER of at least $10^{-3}$ over the full communication chain with an external side. BPSK is used for the downlink because it has a better theoretical bit error rate performance than other modulation schemes, such as amplitude shift keying and frequency shift keying [22]. The BER target is checked by using the IC results in the overall link budget. The focus of this work is on a low-power implanted side transceiver IC. Any need for improved BER could be taken care of by adjusting the link budget through the coil design with the external side. The case of the uplink, for example, where the IR-UWB receiver is external, reported receivers [23,24] at similar sensitivities already meet the BER target of $10^{-3}$. Similarly, for the downlink case, where the transmitter is external, the transmit voltage can be scaled easily to meet the implanted receiver sensitivity of 50 mV. In addition, straightforward coil design can already realize the required channel bandwidth.

4.  Security: with the increasing risk of communication security breaches, the wireless link needs to be secure, especially at the physical layer. Therefore, short range transcutaneous communication is proposed from beneath the skin to the receiver just outside the head. The expected transmission path through skin is expected to be in the range of 3–7 mm [25].

5.   Co-existence with other sub-systems: in the overall wireless system of the visual prosthesis, the downlink and wireless powering are also present. The wireless link should be able to cope with other sub-systems in terms of frequency spectrum use, interference, and cross talk. Using the 3–5 GHz band for the uplink and the 4–12 MHz band for downlink provides sufficient frequency spacing to avoid interference. In addition, we propose the use of a coupled inductive link. Furthermore, the lower 3–5 GHz band of the 3–10 GHz for UWB is preferred due to lower attenuation through the skin [9]. In principle, pulse-based systems like IR-UWB are carrierless, and our target frequency band is 3–5 GHz as opposed to a carrier frequency, which is about 4–12 MHz for the BPSK communication in the downlink. Interference will be minimal due to its localization and near-field nature.

The above considerations and requirements are summarized in Table 1.

**Table 1.** Summary of system requirements.

|  | Uplink Transmitter | Downlink Receiver |
|---|---|---|
| Power conusumption | <10 mW | <10 mW |
| Data rate | >23 Mbps | >200 kbps |
| Bit error rate | $<10^{-3}$ | $<10^{-3}$ |
| Security | 3–7 mm short link | 3–7 mm short link |
| Frequency band | 3–5 GHz | 4–12 MHz |

## 3. Implanted Transceiver

### 3.1. IR-UWB Transmitter

To achieve the low power consumption, the IR-UWB transmitter can be designed as a radio frequency integrated circuit (RFIC) rather than with discrete microwave components [26]. The IR-UWB transmitter can be implemented as an RFIC using CMOS technology. Since the transmit power of IR-UWB is low due to the FCC restriction [27], a simple CMOS IC is sufficient. This renders the CMOS RFIC transmitter to consume little power and it allows for integration with the downlink implanted receiver in the same IC.

Figure 2 shows the overall circuit diagram of the proposed CMOS IC IR-UWB transmitter. The IR-UWB transmitter is comprised mainly of the modulator, the impulse generator and a current-controlled oscillator. The design of the transmit antenna, which will be connected to the chip, should be compact and wide band. Although this is also an important aspect for the overall system, it is outside the scope of this paper. To attain low power consumption and reduce complexity, an on-off keying modulation (OOK) scheme is used. It modulates the short pulses (impulse signals) that come from the impulse generator. The modulator can be implemented in CMOS technology by a simple D-Latch [28], making it a digital component. The data bits and the current-controlled oscillator clock are fed as the control signal and D-input of the latch, respectively. The resulting signal is fed to the impulse generator (see Figure 2).

Implementing the impulse generator in CMOS technology can be done with a tunable delay element, followed by squaring of both the signal and its delayed version, and then a NOR-logic gate to form the pulses. The oscillator used for pulse generation in the pulse generator is a five-stage current-starved single-ended ring oscillator as reported in [29]. Figure 3 shows the circuit schematic of the five-stage current-starved single-ended ring oscillator. The difference compared to [29], however, is that five stages was used in our work, and an extra transistor for current control makes it differ from [29].

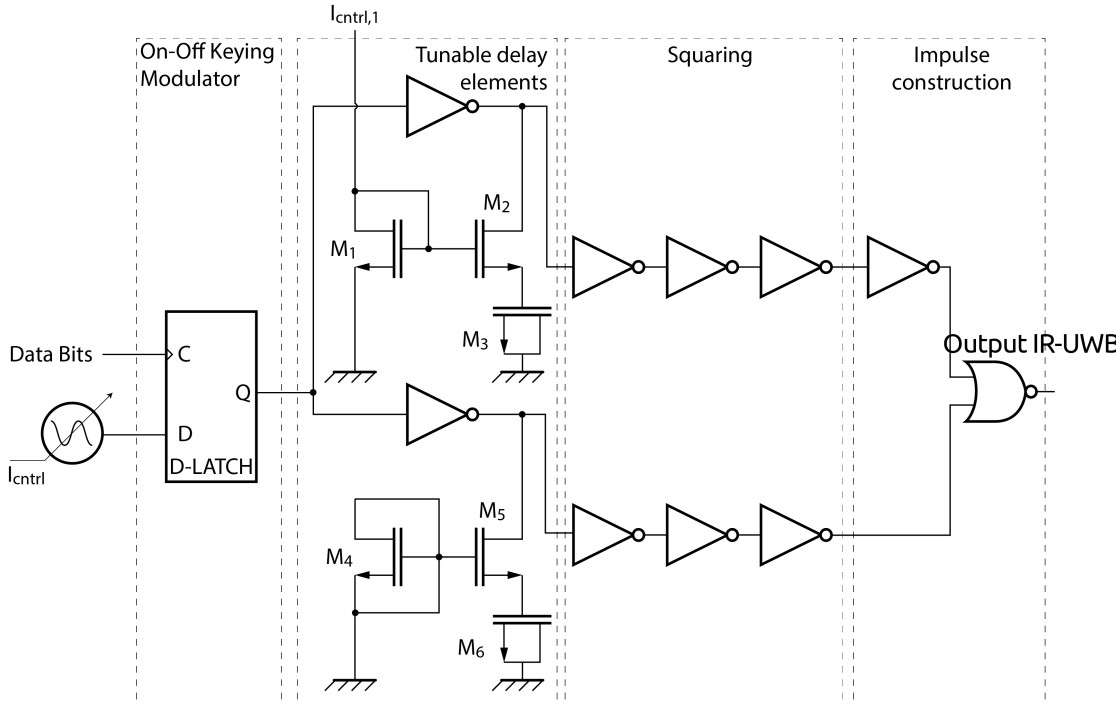

**Figure 2.** Schematic of the IR-UWB transmitter. Note that the presented and measured IC does not include the filter nor the antenna.

For tunable delay elements, a current-controlled shunt-capacitor as in [30] is proposed. Tuning is realised via transistors M1, M2 and M3, while transistors M4, M5 and M6 are used for balance, see Figure 2. Transistor M1 transforms the control current into voltage and biases M2. By changing the input current, the resistance of M2 is tuned. This changes the capacitive loading of M3 on the first stage of the inverter chain. The tunable delay elements add flexibility and help shaping the spectrum of the impulse signals as first explored in [31], our differs with an extra transistor to make it current-controlled rather than voltage-controlled. For proper impulse forming using a NOR-logic gate, the squareness of the signals coming from the tunable delay lines is crucial, and this is done by using three inverters on both the reference line and the delay line for symmetry. Figure 4 shows the IC layout design of the IR-UWB transmitter. From the figure it can be seen that the NOR gate, which is the impulse construction block, is made large enough to drive an antenna that could be connected to it in future work. Large transistor sizes improve the drive capability of the last stage, especially when logic gates are used [32].

Finally, pulse shaping filters are needed, which can be implemented off-chip either by relying on the transmission parameter (S21) from the biological tissue (skin) between the transmit/receive antenna or by a bandpass filter. The shape response is similar if they are designed to pass the same band, and does not impact non-coherent detection of the OOK IR-UWB signal at the external side in our 3–7 mm near-field scenario. In open air, far-field transmission, pulse shaping filters are relevant to make the system satisfy the FCC mask. However, in our context, with transmission through 3–7 mm of skin, already 20–25 dB loss is seen, which causes the radiated power at the surface of the head to be already far below the FCC mask. This is based on our transmitter delivering −27 dBm and the aforementioned attenuation through skin. In spite of these losses, the link budget can still be met.



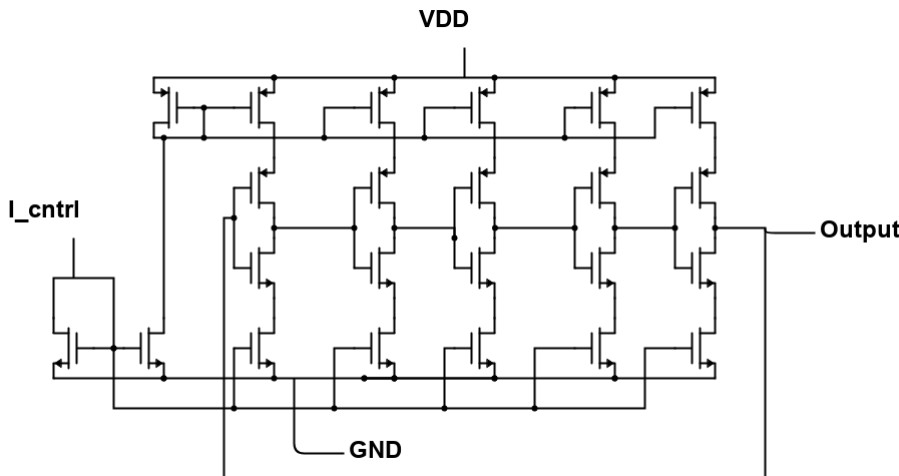

**Figure 3.** Schematic diagram of the 5-stage current-starved single-ended ring oscillator for receiver and transmitter.

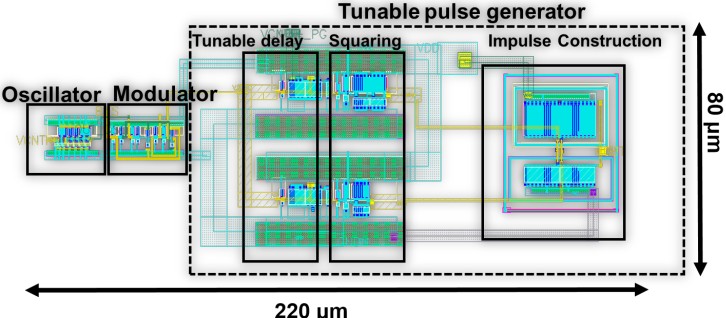

**Figure 4.** IC layout of the IR-UWB transmitter.

*3.2. Non-Coherent BPSK Receiver*

Similarly to the transmitter design for uplink, the requirements for downlink can be met by taking a system approach aimed at designing a low-power implementation. This involves using low frequency (<100 MHz), an inductive link, phase shift keying (PSK), and a nearly digital receiver with non-coherent demodulation after sampling and by means of edge detection, as introduced in [19,33,34]. The sampling of the received signal makes it possible for a non-coherent demodulation in the BPSK signal by using only edge detection, as described in [35]. We say the receiver is nearly digital because of the use of predominately digital components.

Figure 5 shows the block diagram of the receiver. At the receiver side, the entire signal is sampled. The sampling is essentially done using a comparator, sometimes also called a 1-bit analog to digital converter (ADC). This comparator is a dynamic comparator as in [36]. Its threshold, which is the input common mode voltage, can be set by Vcntrl comparator, as shown in Figure 5, and the sensitivity is found to be below 50 mV (in simulations). The comparator shares the same clock with the non-coherent digital demodulator. Figure 6 shows the circuit schematic of the 1-bit ADC which is based on [36]. After this 1-bit ADC stage, the resulting signal is non-coherently digitally demodulated. The non-coherent digital demodulator is the central part of the receiver, and it will be detailed next.

Figure 7 shows the schematic of the non-coherent-digital demodulator. The non-coherent digital demodulator detects if a '0' or a '1' was transmitted by detecting the type of edge it encounters in the digitized received modulated signal. Note that the digitized modulated signal has a falling edge for the '0' and a rising edge for the '1'. While detecting which type of edge is present, the sub-system must take care to avoid the transition points between symbols so that these are not detected as edge types. It is comprised mainly of

an edge detector, a reset generator, and an oscillator. A similar concept was presented in [35], but in our case, the data rate to carrier frequency ratio is not 100% due to practical bandwidth limitations and losses in inductive links. Inductive coils that are designed for transmission and reception create a bandwidth for which the BPSK signal is communicated, and this is band limited by design which leads to about 5–70% data rate to carrier frequency ratio in practice [34]. We aim to be within this range, as delivering a sufficient data rate is the main priority, and because the carrier frequency can be scaled appropriately.

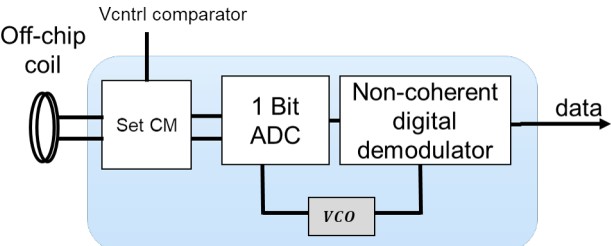

**Figure 5.** Schematic diagram of the BPSK receiver.Note: IC excludes the coil.

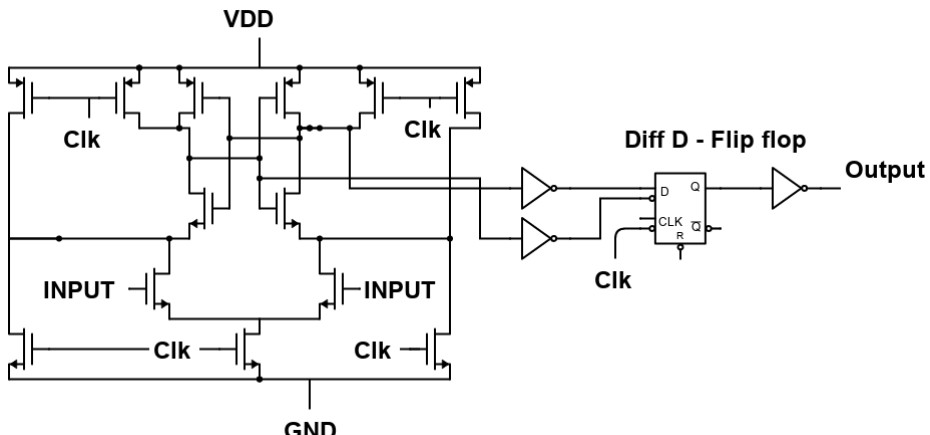

**Figure 6.** Schematic diagram of the 1-bit ADC.

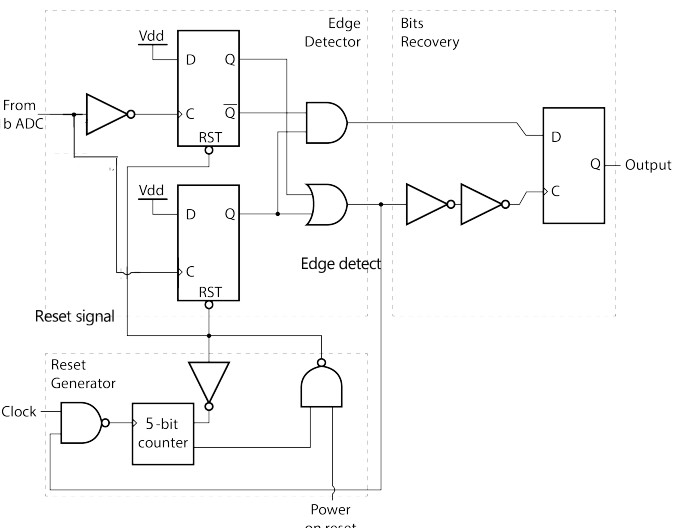

**Figure 7.** Schematic of non-coherent digital demodulator.

The edge detector uses a rising edge D-type flip-flop and a falling edge D-type flip-flop to detect if an edge occurs and which type of edge it is, using an OR logic gate and an AND logic gate, respectively. These logic gates will take the outputs of the flip-flops as their input. The inputs of both flip-flops are a logical '1' while the received digitized modulated

signal is fed as its clock input. This configuration yields the desired effect of edge detection in a power efficient way. The asynchronous clear input of the flip-flops is connected to the reset generator. Asynchronous clearing implies that the flip-flops are cleared decoupled from clock ticks. This is done to reset the flip-flops to avoid detecting the transition point between symbols.

The role of the reset generator module is to provide a reset signal to reset the edge detector at a time after an edge detection at $t = 0$, between $t = 0.5T_{PSK}$ and $t = T_{PSK}$. Here, $T_{PSK}$ is the period of the carrier frequency. To achieve this, it counts, using an asynchronous counter. We determine the frequency of the clock, which is essentially the frequency of the oscillator, as follows.

Let $N$ be the count-up value $(0, 1, \ldots, N)$. The following inequalities must be satisfied. To reset after the transition point between carrier symbols, we have

$$(N - 1)T_{OSC} > 0.5T_{PSK}. \tag{1}$$

To ensure that there is no reset after the next edge, we set

$$NT_{OSC} < T_{PSK}. \tag{2}$$

Combining (1) and (2), we arrive at

$$\frac{1}{2(N - 1)}T_{PSK} < T_{OSC} < \frac{1}{N}T_{PSK}. \tag{3}$$

Taking reciprocals, the range of the frequency of the oscillator to avoid transition between symbols then results:

$$Nf_{PSK} < f_{OSC} < 2(N - 1)f_{PSK}. \tag{4}$$

A low count-up value of $N$, for example $N = 3$, yields an oscillator frequency range of $3f_{PSK} < f_{OSC} < 4f_{PSK}$. This already strongly relaxes the jitter or quality factor requirement of the oscillator. Thus, a low-power current-controlled oscillator can be used. Similar to the IR-UWB tranmsitter, the oscillator used for clock is in the receiver is another five-stage current-starved single-ended ring oscillator as in Figure 3. Figure 8 shows the layout of the receiver on IC. A count-up number of $N = 16$ was used to improve flexibility of adjusting the timing range, which corresponds to the most-significant bit of the 5-bit counter in the reset generator. More details on the components and principle of operation can be found in our previous works in [21,34].

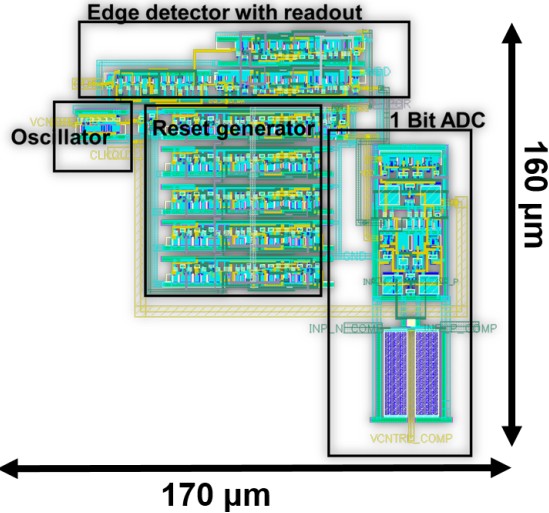

**Figure 8.** IC layout of the BPSK receiver.

## 4. Measurement Setup

### 4.1. Test IC

The IR-UWB transmitter and the BPSK receiver were fabricated on the same die; 1.66 mm by 1.66 mm of the CMOS 180 nm semiconductor technology process. It is a mature and robust technology which is important for medical applications. Figure 9 shows the silicon die. The active areas of the IR-UWB transmitter and BPSK receiver are 220 μm by 80 μm and 170 μm by 160 μm, respectively. Decoupling capacitors were placed to fill the empty area in the die and to improve the signal performance. The die was package in a QFN32 5 mm by 5 mm package. Care was taken to minimize the bond-wire distance for the IR-UWB transmitter by moving it closer to the package pins.

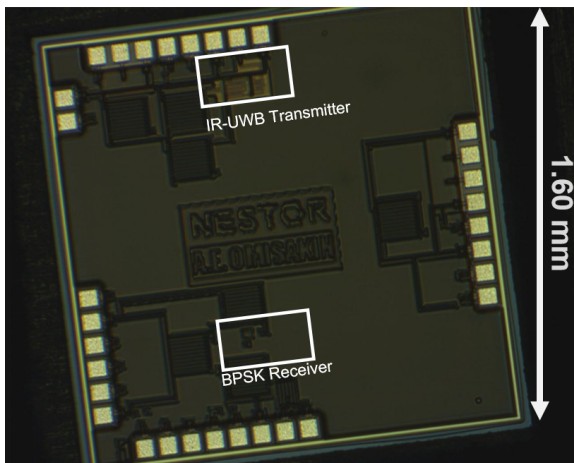

**Figure 9.** Micro-graph of the fabricated transceiver IC.

### 4.2. Demonstrator Board

To test the chip, a printed circuit board (PCB) was fabricated and the packaged chip was mounted on the PCB. The picture inset in Figure 10 shows the manufactured PCB. The board is 84 mm by 95 mm. For alternative test paths and for circuit debugging purposes, more SMAs were included on the PCB. Two possible pathways are made for current control: direct current supply and a potentiometer connected to the voltage supply. These controls are used for the oscillator of the BPSK receiver, the oscillator of the IR-UWB transmitter, and the pulse delay. The IR-UWB transmitter output SMA connector was placed very close to the chip as it is the highest frequency on the PCB.

### 4.3. Experimental Test Setup

Figure 10 shows the block diagram of experimental test setup for the IR-UWB transmitter. The IR-UWB transmitter was tested using the Analog Discovery 2 [37] to generate a bit-stream to be transmitted. Current sources were used to control the oscillator and impulse generator. The LeCroy WaveMaster 830 Zi oscilloscope [38], was used to measure the output of the transmitter signal in time-domain waveform as well as its spectrum.

Figure 11 shows the block diagram of experimental test setup for the IR-UWB transmitter. The BPSK receiver was measured using the Analog Discovery 2 as pattern generator and as an oscilloscope for the low-frequency <100 MHz measurement. A voltage supply was used for powering the circuit and power-on-reset (POR). A current supply was used for oscillator control. Another voltage supply for control voltage for the comparator. A BPSK transmitter was used to generate the modulated signal on 4 MHz carrier with a 1 Mbps data rate.

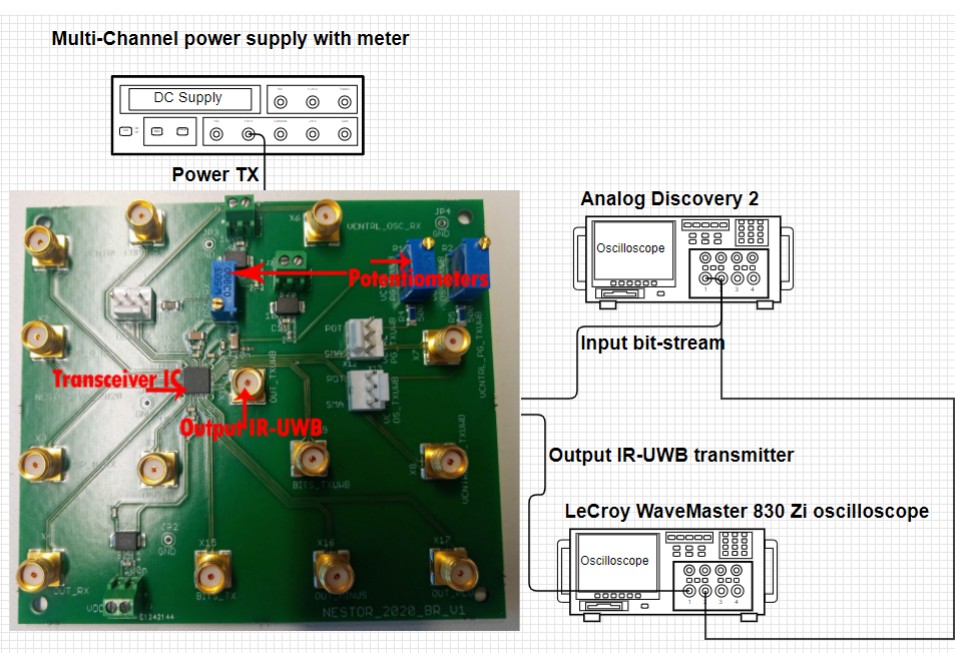

**Figure 10.** Block diagram of experimental test setup for the IR-UWB transmitter with the demonstrator board as picture, the rest of the setup as a diagram.

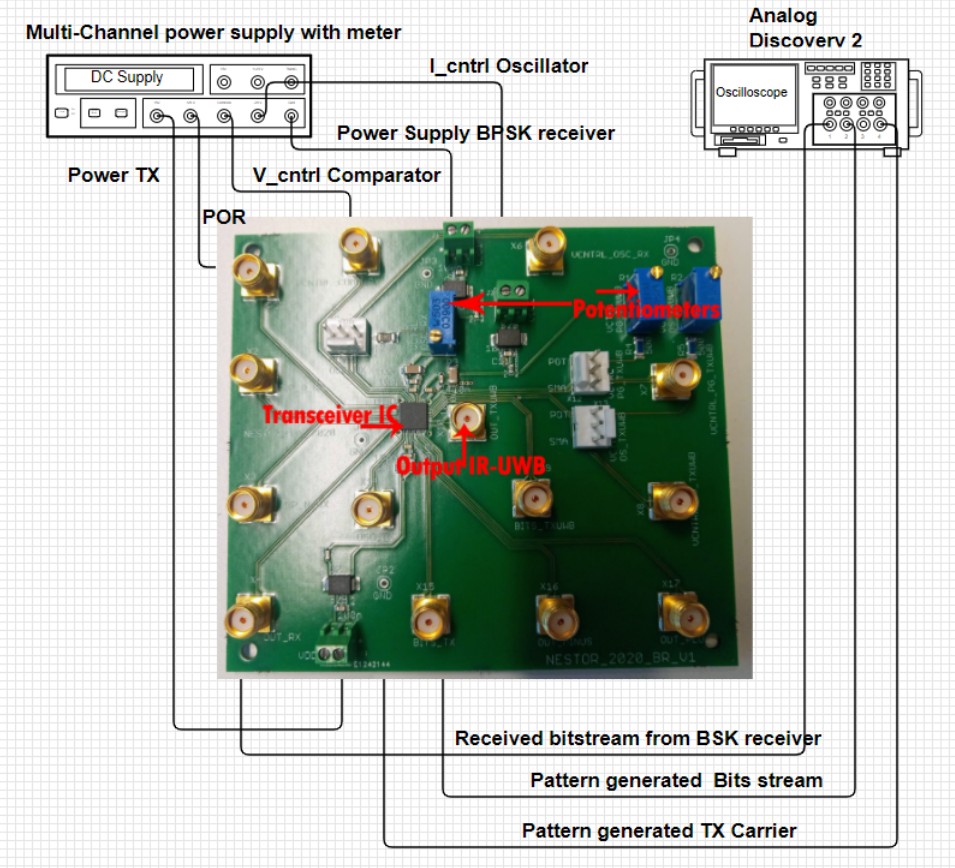

**Figure 11.** Block diagram of experimental test setup for the BPSK receiver.

## 5. Results

In this section, we will show the relevant measurement results for the IC, for both the uplink and downlink subsystems. Furthermore, to show the performance of the subsystem in the context of the overall system requirements, link budgets for the two subsystems are also given.

### 5.1. IR-UWB Transmitter

5.1.1. IC Measurement Results

Figure 12 shows more transmitter output (bottom panel) on a bitstream and shows the input 50 Mbps data stream from the Analog Discovery 2 acting as a pattern generator (top panel). The measured transmitted bits appear to be distorted due to the bandwidth limitation of the pattern generator and also due to the distortion at the interface when the input bit signal is tapped into the oscilloscope, which does not influence performance. A '1' logic level corresponds to no pulse on the output, while a '0' logic level corresponds to a pulse at the output. This demonstrates the on-off keying modulation. Although not depicted, the output voltage levels of the measured and simulated signals agree.

Figure 13 shows the spectrum (FFT of the time signal) of the transmitter output and also with and external bandpass filter for the 3–5 GHz band. To show the additional effect of a bandpass filter, the VBFZ-3590-S+ from Mini-circuits USA, which is a bulky off-chip BPF, is included in a measurement. The measured output spectrum falls below the FCC spectral mask limits. The transmitter waveform will further meet the IR-UWB standard in a full system scenario due to the skin attenuation.

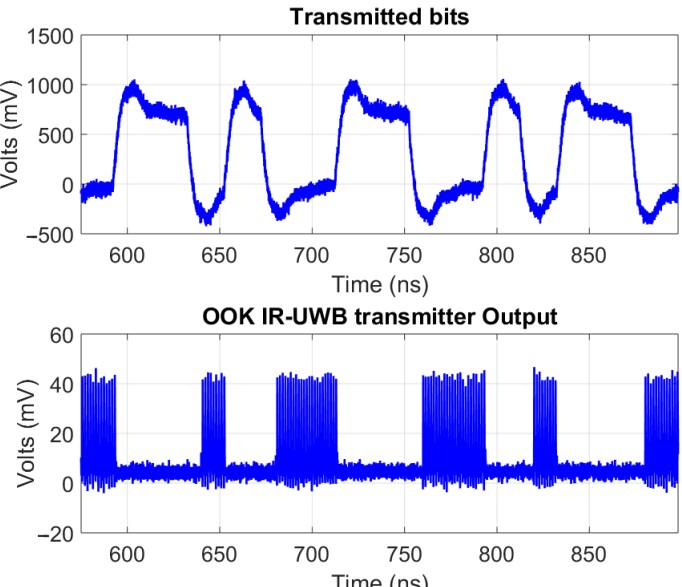

**Figure 12.** Measured time domain waveform of the IR-UWB transmitter at 50 Mbps bitstream, 1.3 V supply voltage. The top panel shows the input bitstream. The bottom panel shows the output of the transmitter IC, which is the modulated data. The measurement has been done without antennas for the sake of convenience.

Figure 14 shows the measured power consumption of the transmitter versus supply voltage. This is not the optimized result yet, and is not able to be able to compared with the post-layout simulation. A 1 kΩ resistance from a potentiometer in series with the supply voltage was used to generate current for the oscillator's current control. Another 1 kΩ resistance from a potentiometer in series with the supply voltage was used to generate current for the impulse generator's current control. The measured and simulated power consumption agree well and show the same trend. Figure 15 shows the corresponding simulated RF output power versus voltage supply for the unoptimized IR-UWB Transmitter

at 50 Mbps data rate vs supply voltage, with fixed 1 kΩ resistance for current control. Another purpose of Figure 14 is to show how the power consumption scales with the supply voltage to emphasise the predominantly digital architecture. This gives a promise for even better performance with more advanced CMOS technologies, operating operates at lower power supply voltages. The results in Figure 14 are for comparison only and are not optimized yet. The optimization was done as shown below.

The current from both current supplies (to control the oscillator and pulse generator) were adjusted to find optimal pulses per bit and power-output level. At 1.3 V supply voltage, using 100 µA and 10 µA for impulse generator and oscillator control respectively, the transmitter consumed 0.15–0.3 mW. The output waveform for this setting, already shown in Figure 12, has an RMS output power at the transmitter of −27 dBm. The simulated power breakdown of the transmitter is as follows: 75% is consumed in the pulse generator, 20% in the oscillator, and 5% in the modulator.

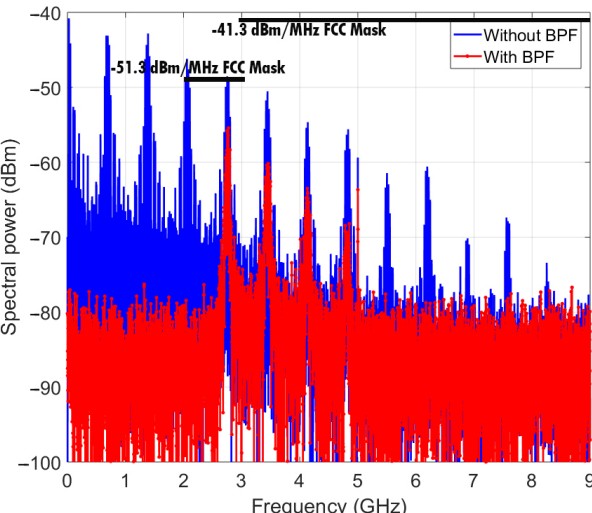

**Figure 13.** Spectrum of the IR-UWB transmitter before and after bandpass filter at 1.3 V supply voltage. The FCC spectral mask is also shown. The spectrum is computed in the oscilloscope as the FFT of the measured time signal.

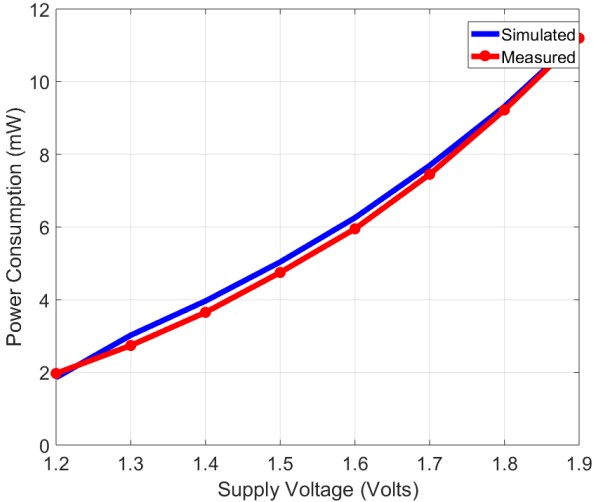

**Figure 14.** Power-consumption versus voltage supply for the unoptimized IR-UWB Transmitter at 50 Mbps data rate with fixed 1 kΩ resistance for current control.

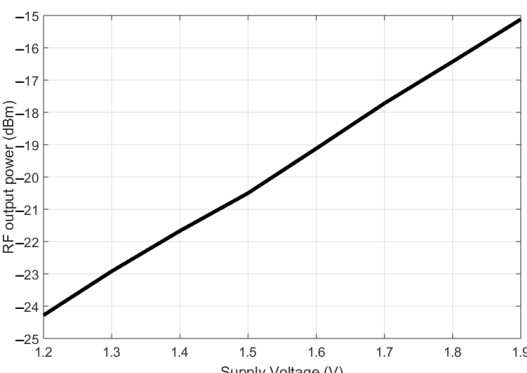

**Figure 15.** RF power versus voltage supply for the unoptimized IR-UWB Transmitter at 50 Mbps data rate vs supply voltage, with fixed 1 kΩ resistance for current control.

### 5.1.2. Link Budget for Uplink

It is important to analyze how the transmitter would fit in the complete chain, with a transmit/receive antenna pair, communication through 3–7 mm of skin, and an external receiver. For this purpose, we look at the link budget. Our IR-UWB transmitter delivers −27 dBm. The FCC regulations impose approximately a −9 dBm limit for a 2 GHz bandwidth in the UWB 3–10 GHz spectral mask of −41.3 dBm/MHz. For the −27 dBm output, the consumed power is 0.15–0.3 mW on 1.3 V supply. The S21/link loss through 3 mm skin is found to be approximately 20 dB from electromagnetic simulations (the worst-case scenario at 7 mm skin thickness gives an extra 5 dB loss around the target 3–5 GHz band), which means that −47 dBm will reach the LNA of the external receiver or −52 dBm at 7 mm skin thickness. This seems well within reach. A similar antenna skin interface design was shown in [9], which agrees with our simulated loss. The simulation was carried out in SIMULIA Studio Suite with antenna dimensions of 12 mm by 12 mm, thickness of 1 mm and insulator coating of 1 mm. Reported UWB receiver IC sensitivies are about −70 dBm to −65 dBm [23,24,39], which implies that we have an excess margin of 18–23 dB for our application at 3–7 mm assuming a −70 dBm sensitivity of the external receiver. BER performance assumes that, at this sensitivity, the $<10^{-3}$ target is reached [23,24], otherwise the sensitivity is lowered, taking into account the 18–23 dB excess margin. Finally, error detection and protection methods such as linear and convolutional codes could further increase the link margin which is already in excess.

### *5.2. Non-Coherent BPSK Receiver*

#### 5.2.1. IC Measurement Results

We connect directly without coils delivering a 200 mV modulated signal level into the receiver to examine its power consumption. Figure 16 shows the transmitted input bits and the recovered output for the receiver. These are recorded using the following settings: 1.3 V supply, a Vcntrl comparator setting of 0.6 V (the input common mode voltage level), and a current source for oscillator controlled at 9.3 μA. To better illustrate internal signal of the receiver, Figure 17 shows the internal timing diagram from simulation, in which the demodulation of the received signal can be observed. The BPSK receiver consumed 0.2 mW while demodulating 1 Mbps data rate which was on a 4 MHz carrier. Figure 18 shows the measured power consumption of the receiver versus supply voltage at a 1 MHz repeated square wave cycle. The purpose of Figure 18 is to show the dependence of the power consumption on the carrier frequency and the supply voltage. A fixed repeated data square wave cycle of 1 MHz has two times the rate of a 1 Mbps bit-stream and should have a higher power consumption than the reported 0.2 mW at 1 Mbps. The decrease in power consumption as the supply voltage is lowered can be observed. The higher power consumption at a 5 MHz carrier demonstrates the predominately digital architecture. The simulated power breakdown of the receiver is as follows: 35% is consumed in the 1-bit ADC, 35% in the digital demodulator, and 30% in the oscillator(clock).

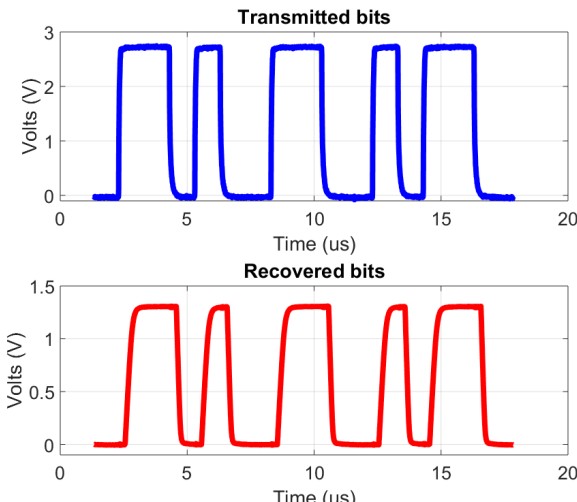

**Figure 16.** Measured BPSK receiver recovered bits on 1 Mbps on 4 MHz carrier.

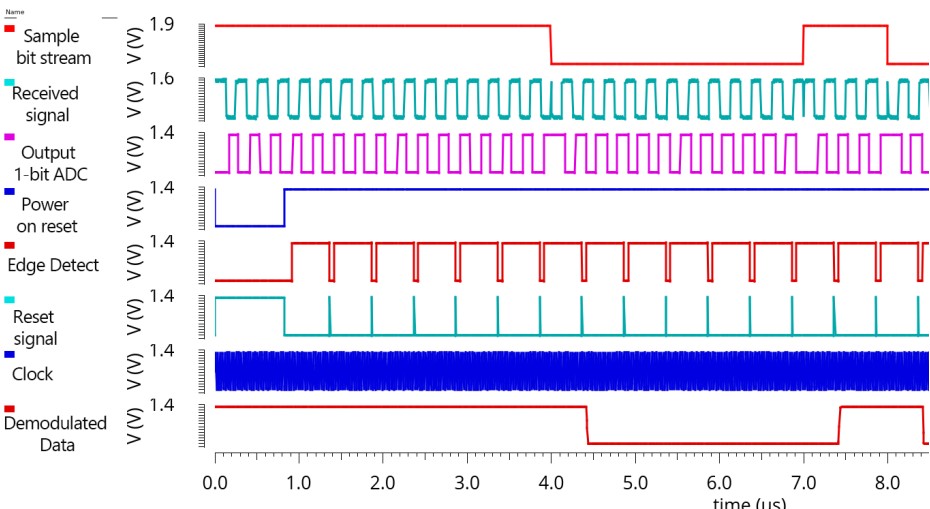

**Figure 17.** Simulated waveforms and timing diagram of the BPSK receiver.

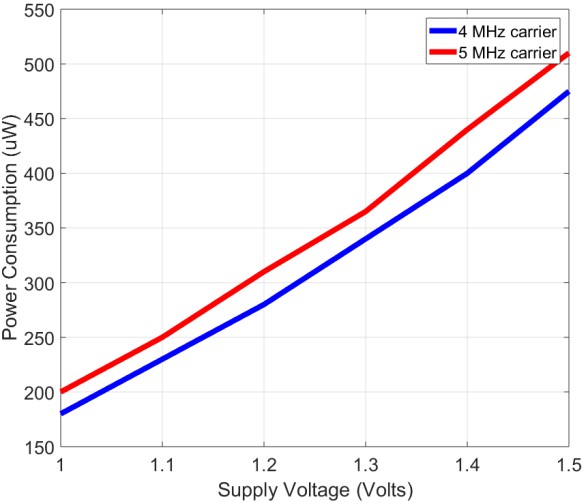

**Figure 18.** Measured power consumption of the receiver vs supply voltage at 1 MHz repeated data square wave cycle.

5.2.2. Link Budget for Downlink

Similarly to the uplink case, it is important to analyze in a link budget how the receiver would fit with the transmit/receive coils, while taking into account 3–7 mm transcutaneous communication with the transmitter on the outside. The BPSK receiver sensitivity is about 10–50 mV, determined by its comparator, at the first stage. Furthermore, from our coil and transmitter simulation, a voltage level of 200–2500 mV could reach the receiver, depending on the coil design and distance [21]. Typical coil sizes range from 15 mm to 40 mm in diameter depending on the size desired. The coupled inductive link from the coils acts as a low-power voltage transformer, boosting the voltage around resonance frequency it is designed for [34]. Typical coil dimensions range from 10 mm to 50 mm in diameter to ensure that the coil self-resonance frequency is much higher than our operating frequency. In addition, these coils sizes result in a tolerable misalignment of several mm. However, here we connect directly without coils to deliver a 200 mV level into the receiver. From a sensitivity of 50 mV and a possible coil output of 200–2500 mV, the link will be closed with a 12–26 dB excess margin for downlink when the coupled coil response has the desired bandwidth of about 2 MHz, as investigated in [21]. BER performance assumes that the $<10^{-3}$ target is reached; otherwise, the receiver power level can be adjusted from the external transmitter and coil design, considering the 12–26 dB excess margin. Finally, error detection and protection methods such as linear and convolutional codes could further increase the link margin.

## 6. Discussion

### 6.1. Comparison with the State of the Art

Next, we benchmark our work with state of the art results from the literature. Table 2 compares our implanted transmitter with recently published general transmitter ICs. Clearly our transmitter performs similarly, or better, in terms of data rate. Power consumption is better than reported in [40–42] with the standard 180 nm CMOS technology available for IC design in our lab, but worse than reported in [43,44]. The latter two [43,44], however, use 0–200 MHz and use the body as a wire, respectively. The systems in [43,44] are not suitable to meet the co-existence requirements for our overall system application. This is mainly due to differences in applications and intended ranges. However, this stresses the need for the low-power approach that we followed for the intracortical visual prosthesis.

Table 3 compares with other high data rate state-of-the art medical implants. We achieved a lower power consumption but at a lower data rate, which is still sufficient for our current application. This shows that our IR-UWB transmitter approach to use predominantly digital components and simple architecture is effective. In [45–47], a much lower data rate of <30 Mbps was reported, which is not well-suited for our data rate requirements for the uplink. In [48], a higher data rate is shown at slightly lower power consumption, but the higher frequency band of 6.8–9 GHz reportedly used will face about 10 dB more attenuation through the skin tissue.

Table 4 compares the implanted receiver with recently published general receiver ICs. While we have a comparable data rate, we achieve much lower power consumption, except for the results reported in [43] which is not designed for skin tissue interaction, its frequency spectrum of 0–200 MHz may not co-exist in frequency band with other parts of our system.This due to our short-range and coupled-coil link, which relaxes the need for very sensitive receiver, which would have raised the power consumption.

Table 5 compares our receiver with other medical implants. Our receiver is of comparable data rate with much lower power consumption. Our power consumption is better than in the first five columns and only worse than in [49]. However, the data rate of 0.01 Mbps reported in [49]  is low and 20 times below the minimum data rate requirement of 0.2 Mbps for the intracortical visual prosthesis. This shows that our non-coherent digital demodulator is effective in low-power consumption compared to traditional receivers.

**Table 2.** Comparison with recently published IC-transmitter. (HBC—Human body communication, TTC—Transmission time control).

|  | [40] | [41] | [42] | [43] | [44] | This Work |
|---|---|---|---|---|---|---|
| Data rate | 1 Mbps | 1 Mbps | 11 Mbps | 50 Mbps | 30 Mbps | 50 Mbps |
| Power consumption | 3.7 mW | 0.42 mW | 4 mW | 0.0237 mW | 0.093 mW | 0.3 mW |
| Frequency | 2.4 GHz | 3–5 GHz | 401–428 MHz | 0–200 MHz | Broadband | 3–5 GHz |
| Modulation | BLE | UWB | QPSK | TTC | HBC | UWB |
| Technology | 28 nm | 180 nm | 130 nm | 180 nm | 180 nm | 180 nm |
| Supply Voltage | 1 V | 1.8 V | 1 V | 1 V | 1 V | 1.3 V |

**Table 3.** Comparison with other medical implant transmitters.

|  | [10] | [11] | [12] | [48] | [45] | [47] | [46] | This Work |
|---|---|---|---|---|---|---|---|---|
| Data rate | 100 Mbps | 90 Mbps | 500 Mbps | 100 Mbps | 3 Mbps | 30 Mbps | 0.14 Mbps | 50 Mbps |
| Power consumption | 2.1 mW | 1.6 mW | 5.4 mW | 0.26 mW | 5.4 mW | 30 mW | 0.085 mW | 0.3 mW |
| Frequency | Light | 3–5 GHz | 3–7 GHz | 6.8–9 GHz | Sub-GHz | Sub-GHz | 3–5 GHz | 3–5 GHz |
| Modulation | - | UWB | UWB | UWB | UWB | UWB | UWB | UWB |
| Technology | - | 350 nm | 130 nm | 180 nm | 180 nm | 350 nm | 90 nm | 180 nm |
| Supply Voltage | - | 1.65 V | 1.8 V | 1.5 V | - | 3.3 V | - | 1.3 V |

**Table 4.** Comparison with recently published IC-receivers. (DDM—Differential detection method, BLE- Bluetooth Low Energy).

|  | [50] | [51] | [52] | [53] | [54] | [55] | [43] | This Work |
|---|---|---|---|---|---|---|---|---|
| Data rate | 1.3 Mbps | 10 Mbps | 11 Mbps | 1 Mbps | 1 Mbps | 1 Mbps | 50 Mbps | 1 Mbps |
| Power consumption | 5.2 mW | 3.2 mW | 2.4 mW | 9.6 mW | 0.9 mW | 1.5 mW | 0.041 mW | 0.2 mW |
| Frequency | 18–23 MHz | 40–120 MHz | 2.4 GHz | 2.4 GHz | 2.4 GHz | 2.4 GHz | 0–200 MHz | 4 MHz |
| Modulation | BPSK | Double FSK | BLE | BLE | BLE | BLE | DDM | BPSK |
| Technology | 130 nm | 180 nm | 40 nm | 65 nm | 28 nm | 28 nm | 180 nm | 180 nm |
| Supply Voltage | 1.2 V | 1 V | 1 V | 3 V | 0.7 V | 0.7 V | 1 V | 1.3 V |

**Table 5.** Comparison with other medical implant receivers. (P-OFDM—Pseudo orthogonal frequency-division multiplexing).

|  | [16] | [8] | [17] | [18] | [56] | [49] | This Work |
|---|---|---|---|---|---|---|---|
| Data rate | 2 Mbps | 100 kbps | 13.56 Mbps | 8 Mbps | 2 Mbps | 0.01 Mbps | 1 Mbps |
| Power consumption | 6.2 mW | - | 2.2 mW | 0.6 mW | 1.1 mW | 0.092 mW | 0.2 mW |
| Frequency | 20 MHz | 5 MHz | 13.56 MHz | 902–928 MHz | 20–120 MHz | 413–419 MHz | 4 MHz |
| Modulation | DPSK | - | PDM | FSK-ASK | P-OFDM | OOK | BPSK |
| Technology | 350 nm | - | 350 nm | 130 nm | 65 nm | 180 nm | 180 nm |
| Supply Voltage | - | 1 V | 1 V | 3 V | 1.1 V | 1.5 V | 1.3 V |

### 6.2. Medical Safety

It is essential to mention the safety of the system with regards to the SAR limits and tissue heating. Our implanted transceiver has sub-milliWatt power consumption. In [57]

1000 mW was transmitted in SAR simulations resulting in 10 g averaged SAR levels of a maximum of 1.92 W/kg are obtained. This value is below the specified 2 W/kg safety level. With the total power consumption of our IC below 1 mW, the data communication through the skin will be far below the SAR limit. The implanted transceiver itself can also dissipate power, causing the tissue directly surrounding it to heat up. The maximum temperature increase in the cortex has to be smaller than 1 °C [58,59]. This corresponds to a maximum power density of 0.8 mW/mm$^2$ of exposed tissue area [20,58,59]. Our measured implanted transceiver IC consumes 0.5 mW in a 25 mm$^2$ QFN package, which brings the power density to 0.02 mW/mm$^2$, which is well below the safety limit. In [60], research into electromagnetic and thermal effects on IR-UWB on the human head was reported. From this work, it was also confirmed that the power levels for our IC are well within the control of thermal regulatory mechanisms of the human body.

## 7. Conclusions

A sub-milliwatt transceiver IC for the implant side of an intracortical visual prosthesis was designed, fabricated and measured. It delivers 1 Mbps for the downlink (for stimulation) and 50 Mbps for the uplink (for recording), using a non-coherent BPSK demodulator and an IR-UWB transmitter, respectively. Its predominately digital components and adjustability lead to the low power consumption of 0.2 mW for the BPSK receiver and 0.3 mW for the IR-UWB transmitter at 1.3 V supply on 180 nm CMOS technology. Based on our transceiver IC in the implant, the system link budget analysis for both uplink and downlink show achievable figures: there is 18–23 dB excess margin for uplink, and 12–26 dB for downlink. These figures show that the link can be closed with an excess margin for the antenna/coil pair to communicate through 3–7 mm skin, which is the transcutaneous interface between the implant side and outer side, while achieving good BER performance as a result of closing the link with an excess margin.

In the future, scaling to more advanced processes that use lower supply voltages will further drastically reduce power consumption due to the predominately digital architecture. Furthermore, the design of antennas/coils and a transceiver IC on the outside, in order to realize the system link budget, is the logical next step.

**Author Contributions:** Investigation, A.O. and G.R.; Conceptualization and methodology, A.O. and G.R.; Supervision, R.M. and M.B.; Funding acquisition, M.B.; Data curation, A.O.; Writing—original draft, A.O.; and Writing—review and editing, A.O., R.M., G.R. and M.B. All authors have read and agreed to the published version of the manuscript.

**Funding:** This research was funded by the Dutch Technology Foundation STW, which is part of the Netherlands Organization for Scientific Research (NWO), project 5 of the NESTOR program (P15-42).

**Data Availability Statement:** Data is contained within the article tables and figures and is available on request.

**Acknowledgments:** The authors would like to thank Pieter Harpe from the Eindhoven University of Technology for his reviews and support during the IC design and fabrication process. This research is supported by the Dutch Technology Foundation STW, which is part of the Netherlands Organization for Scientific Research (NWO), project 5 of the NESTOR program (P15-42).

**Conflicts of Interest:** The authors declare no conflict of interest.

## Abbreviations

The following abbreviations are used in this manuscript:

| | |
|---|---|
| ISM | Industrial, Scientific, and Medical |
| IC | Integrated Circuit |
| SAR | Specific Absorption Rate |
| NESTOR | NEuronal STimulation for Recovery of Function |
| BPSK | Binary Phase Shift Keying |
| PLL | Phase Locked Loop |

| | |
|---|---|
| FSK | Frequency Shift Keying |
| ASK | Amplitude Shift Keying |
| BER | Bit Error Rate |
| CMOS | Complementary Metal–Oxide–Semiconductor |
| SPST | Single Pole Single Throw |
| IR-UWB | Impulse Radio Ultra Wide Band |
| OOK | On-Off Keying |
| DDM | Differential detection method |
| BLE | Bluetooth Low Energy |
| HBC | Human body communication |
| TTC | Transmission time control |
| P-OFDM | Pseudo orthogonal frequency-division multiplexing |
| RFIC | Radio Frequency Integrated Circuits |
| ADC | Analog to Digital Converter |

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
