# Peer review of "Sub-Milliwatt Transceiver IC for Transcutaneous Communication of an Intracortical Visual Prosthesis"

_electronics, doi:10.3390/electronics11010024_

Round 1
Reviewer 1 Report
This paper is well written and easy to find the difference between the proposed and previous works from the tables. I do not have particular negative feedback for this paper.
Author Response
Comments and Suggestions for Authors
This paper is well written and easy to find the difference between the proposed and previous works from the tables. I do not have particular negative feedback for this paper.
REPLY: Thank you for the positive feedback. It is appreciated.
Reviewer 2 Report
The structural parts of this scientific article are good:
- Introduction(Fig.1 and literature sources 1-17);
- System Requirements and Considerations (Table 1and literature sources up 23);
- Implanted Transceiver (Fig.2-Fig.6 and literature sources up 34);
- Measurement Setup (Fig. 7-Fig. 8 , and literature sources 35-36);
- Results (Fig. 9-Fig. 12 , and literature sources - new 37);
- Discussion (Fig. 13-Fig. 14 , Table 3-5 and literature sources – 38 -58);
Conclusion and References [1-58].
The relevance of the study is substantiated and good experimental results have been obtained.
A sub-milliwatt transceiver IC for the implant side of an intracortical visual prosthesis is designed, fabricated and measured. It delivers 1 Mbps for the downlink (for stimulation) and 50 Mbps for the uplink (for recording), using a non-coherent BPSK demodulator and an IR-UWB transmitter, respectively. Its predominately digital components and adjustability lead to the low power consumption of 0.2 mW for the BPSK receiver and 0.3 mW for the IR-UWB transmitter at 1.3 V supply on 180 nm CMOS 420 technology. Based on our transceiver IC in the implant, the system link budget analysis for both uplink and downlink show achievable figures: there is 18–23 dB excess margin for uplink, and 12–26 dB for downlink.
Author Response
Comments and Suggestions for Authors
The structural parts of this scientific article are good:
Introduction(Fig.1 and literature sources 1-17);
System Requirements and Considerations (Table 1 and literature sources up to 23);
Implanted Transceiver (Fig.2-Fig.6 and literature sources up to 34);
Measurement Setup (Fig. 7-Fig. 8 , and literature sources 35-36);
Results (Fig. 9-Fig. 12 , and literature sources - new 37);
Discussion (Fig. 13-Fig. 14 , Table 3-5 and literature sources – 38 -58);
Conclusion and References [1-58].
REPLY: Thank you for the insightful analysis. It is appreciated.
The relevance of the study is substantiated and good experimental results have been obtained.
A sub-milliwatt transceiver IC for the implant side of an intracortical visual prosthesis is designed, fabricated and measured. It delivers 1 Mbps for the downlink (for stimulation) and 50 Mbps for the uplink (for recording), using a non-coherent BPSK demodulator and an IR-UWB transmitter, respectively. Its predominantely digital components and adjustability lead to the low power consumption of 0.2 mW for the BPSK receiver and 0.3 mW for the IR-UWB transmitter at 1.3 V supply on 180 nm CMOS 420 technology. Based on our transceiver IC in the implant, the system link budget analysis for both uplink and downlink show achievable figures: there is 18–23 dB excess margin for uplink, and 12–26 dB for downlink.
REPLY: Thank you for the positive feedback. It is appreciated.
Reviewer 3 Report
The manuscript presents an UWB transceiver for transcutaneous communication targeting a relatively high bit rate (50Mbps TX, 1Mbps RX). Generally speaking, the power consumption is not ultra-low, but more than satisfactory considering the technology and the bit rate.
Concerning the paper, I have the following remarks:
- the modulator and the demodulator operation need to be better explained. I suggest to add a figure with relevant timing diagrams of the main signals.
- A schematic of the 1b ADC and of the VCOs used in the RX needs to be shown and discussed.
- The chip is tested on board rather than on a full data link. The experimental test setup both for the receiver and for the transmitter needs to be shown in two separate figures (both block diagram and photo)
- Why the input bitstream in Fig.9 is so badly distorted?
- Please report the power breakdown of the different TX and RX blocks. If the measurement is not possible, please report simulated data.
- Please report the RF power corresponding to Fig.11
- Please add more references on digital-based designs (eg:10.1109/RFIT.2011.6141746; 10.1109/TCSII.2021.3049680; )
Author Response
Comments and Suggestions for Authors
The manuscript presents an UWB transceiver for transcutaneous communication targeting a relatively high bit rate (50Mbps TX, 1Mbps RX). Generally speaking, the power consumption is not ultra-low, but more than satisfactory considering the technology and the bit rate.
REPLY: Thank you for your remark.
Concerning the paper, I have the following remarks:
The modulator and the demodulator operation need to be better explained. I suggest to add a figure with relevant timing diagrams of the main signals.
REPLY: Thank you for your remark. We took this into consideration. More details on the modulator and demodulator and further diagrams were already explained in our previous works, cited in the paper to avoid repetition. This paper focuses on the IC validation. We however already showed timing diagrams with Figure 9 and Figure 13 (now 18).
The following text has been added in Section 3.2: “More details on the components and principle of operation can be found in our previous works in \cite{Omisakin2019EMBC, Omisakin2021}”
A schematic of the 1b ADC and of the VCOs used in the RX needs to be shown and discussed.
REPLY: Thank you for the suggestion. The schematics of 1b ADC (a comparator) and VCO have now been added to the paper at the appropriate places.
The chip is tested on board rather than on a full data link. The experimental test setup both for the receiver and for the transmitter needs to be shown in two separate figures (both block diagram and photo)
REPLY: Thank you for your remark. A new subsection: Section 4.3 has been created for this and the block diagram of the experimental setup is shown in separate figures.
In addition, for the reviewer’s curiosity, a link to the video of the measurement setup is here: https://youtu.be/pgDTbRc8lZI. Also, an album of photos of the measurement setup can be found here: https://img.gg/ZZ2YeN6.
Why the input bitstream in Fig.9 is so badly distorted?
REPLY: Thank you for this remark. The distortion in displaying the input bitstream is due to the bandwidth limitation of the Analog Discovery 2 as a pattern generator and also due to the distortion at the interface of the signal transferred to the LeCroy WaveMaster 830 Zi oscilloscope for displaying the signal.
To clarify this to a reader beforehand, the following text has been added in Section 5.1.1: “The measured transmitted bits appear to be distorted, which is caused by the bandwidth limitation of the used pattern generator and also due to the distortion at the interface when the input bit signal is tapped into the oscilloscope which does not influence performance.”
Please report the power breakdown of the different TX and RX blocks. If the measurement is not possible, please report simulated data.
REPLY: Thank you for this remark. This indeed provides more detailed insight into the system performance. The power breakdown is now added in Section 5.1.1 and in Section 5.2.1. “The simulated power breakdown of the transmitter is as follows: 75% is consumed in the pulse generator, 5% in the OOK-modulator and 20% in the oscillator.”
“The simulated power breakdown of the receiver is as follows: 35% is consumed in the 1-bit ADC, 35% in the digital demodulator and 30% in the oscillator (clock).”
Please report the RF power corresponding to Fig.11
REPLY: Thank you for your remark. A new figure has been added that shows the RF output corresponding to Fig.11, and the following text has been added in Section 5.1.1: “Figure 12 shows the corresponding simulated RF output power versus voltage supply for the unoptimized IR-UWB Transmitter at 50 Mbps data rate vs supply voltage, with fixed 1 kΩ resistance for current control.”
Please add more references on digital-based designs (eg:10.1109/RFIT.2011.6141746; 10.1109/TCSII.2021.3049680; )
REPLY: Thank you for the suggestion for these references. These have now been added to the Introduction section as follows: “The potential of digital-based designs is discussed in {Toledo2021,Staszewski2011}, which is beneficial to low-power transceivers.”
Round 2
Reviewer 3 Report
Thank you for addressing my remarks. The quality of the manuscript has improved and it is now ready for publication.